# Ecological Prevalence and Non-Enzymatic Formation of Imidazolium Alkaloids on Moon Snail Egg Collars

**DOI:** 10.3390/molecules31010159

**Published:** 2026-01-01

**Authors:** Karla Piedl, Caitlyn O. Agee, Anthony G. Tarulli, Rose Campbell, Paige Banks, Nicklas W. Buchbinder, R. Thomas Williamson, Emily Mevers

**Affiliations:** 1Department of Chemistry, Virginia Tech, Blacksburg, VA 24061, USA; piedlk89@vt.edu (K.P.); a.tarulli0132@gmail.com (A.G.T.); rosen8@vt.edu (R.C.); banksmp1@gmail.com (P.B.); nwbuch01@vt.edu (N.W.B.); 2Department of Chemistry and Biochemistry, University of North Carolina Wilmington, Wilmington, NC 28403, USA; coa9430@uncw.edu (C.O.A.); williamsonr@uncw.edu (R.T.W.)

**Keywords:** Bacillimidazoles, *Flavobacteriaceae*, marine egg collars, imidazolium

## Abstract

Microorganisms wage constant chemical battles against one another as they compete for space and scarce nutrients, particularly within animal-associated habitats. Here, binary assays were used to investigate chemical interactions among *Flavobacteriaceae* within *Neverita delessertiana* egg collars, a moon snail common to the Gulf Coast. Analysis of 140 distinct pairings revealed eight that exhibited growth-inhibitory activity. Chemical evaluation of the crude extract from *Cellulophaga omnivescoria* EM610, which inhibited the growth of three other *Flavobacteriaceae*, resulted in the isolation of bacillimidazoles A (**1**) and E (**2**), two previously characterized metabolites, isolated from a marine *Bacillus* species. Further work demonstrated that these compounds are readily formed spontaneously by condensation of 2,3-butanedione with phenethylamine and/or tryptamine. Tandem mass spectrometry analysis of the chemical extracts of individual moon snail egg collars revealed the presence of bacillimidazole A in 62% of the egg collars.

## 1. Introduction

Animals depend on bacterial microbiotas that serve essential roles, including defending against predators, producing vital nutrients, and detoxifying natural substrates [1,2,3]. In some cases, these bacterial microbiotas are dominated by a single bacterial species (e.g., fungus-growing ants’ partnership with *Pseudonocardia* sp. and Swinhoe’s sponge partnership with *Candidatus* Entotheonella) [4,5]. In contrast, others have been shown to involve a consortium of diverse bacterial families (e.g., the Hawaiian bobtail squid and amphibian skin) [6,7,8,9]. Some of these interactions are ancient, as evidenced by the reduced genome sizes of the bacterial symbiont and/or the evolution of specialized glands in the animal to house the symbiont [10,11]. Other partnerships are more recent, as supported by population studies [12]. In many of these systems, the bacterial microbiota is predicted to produce small molecules that benefit the animal; in return, the animal provides nutrients that support the growth of the associated bacteria [13]. Recently, several studies have indicated that there is likely fierce competition between the associated bacteria for both nutrients and space within the animal, and the production of specialized small molecules mediates this competition [14,15]. For example, *Acromyrmex* sp., a fungus-growing ant, has been shown to acquire its mutualistic *Pseudonocardia* strain from worker ants within 2 h of eclosion [16]. This has led to antagonism between *Pseudonocardia* strains, where one strain produces a secondary metabolite that inhibits another strain’s growth, even though they are nearly identical, sharing 98.3% sequence identity across chromosomal DNA [17].

Recently, it was discovered that marine egg masses, also known as egg collars, from the moon snail, *Neverita delessertiana*, are enriched with a consortium of bacterial families, dominated by *Flavobacteriaceae* (up to 69%) [18]. *Flavobacteriaceae* are Gram-negative bacteria that play essential roles in nutrient cycling in marine environments by degrading organic matter and pollutants and by establishing biofilms [19,20]. Although *Flavobacteriaceae* have been shown to produce biologically active secondary metabolites, they are considered understudied compared to more-well-studied families, such as Actinobacteria [21,22,23,24]. A recent bioinformatic analysis of *Flavobacteriaceae* genomes indicates they are a rich source of putatively new natural products, containing 2–8 biosynthetic gene clusters (BGCs) per genome, many of which lack hits to known natural products [21,22,25]. To investigate the potential chemical interactions among *Flavobacteriaceae* within the *N. delessertiana* microbiota, a set of binary intruder assays were conducted using an internal library of 36 *Flavobacteriaceae.* These assays identified bacterial strains exhibiting growth-inhibitory activity. Specifically, *Cellulophaga omnivescoria* EM610 inhibited the growth of three *Flavobacteriaceae* strains. Chemical analysis of a *C. omnivescoria* EM610 extract led to the identification and isolation of bacillimidazoles A (**1**) and E (**2**) (Figure 1). Although these compounds are formed spontaneously by the condensation of 2,3-butanedione and biogenic amines, their detection on 62% of the egg collars indicates that they are being produced under ecological conditions.

## 2. Results and Discussion

To investigate the ecological fitness of individual strains of *Flavobacteriaceae* isolated primarily from moon snail egg collars, binary intruder assays were conducted to determine whether these strains could inhibit the growth of other *Flavobacteriaceae*. In these competition assays, a bacterial colony (the “resident”) is inoculated at the center of the Petri dish and allowed to grow until a dense colony forms. Next, a second bacterial strain (the “intruder”) is inoculated at low density around the resident strain and monitored for zones of inhibition and/or morphological changes between the resident and intruder strains. Analysis of 140 distinct pairings revealed eight pairings that yielded a zone of inhibition (Appendix A and Appendix A). Three of these active pairings involved *C. omnivescoria* EM610 as the resident strain, indicating that it was constitutively producing a bioactive metabolite that impacted the growth of *Olleya* sp. EM584, *Sufflavibacter* sp. EM538, and *Zunongwangia* sp. EM537.

A small-scale (1 L) *C. omnivescoria* EM610 culture was grown with hydrophobic resins (HP20, XAD4, and XAD7) to capture secreted small molecules to begin to identify the entity responsible for the growth-inhibitory activity. At 5 d, the resin and cell material were filtered and extracted with an organic solvent to yield a crude extract. The crude extract was initially processed over a solid phase extraction (SPE) C18 column to yield three semi-crude fractions. Spot on lawn assays against *Sufflavibacter* sp. EM538 revealed that the non-polar (100% MeOH) fraction exhibited modest growth inhibitory activity (Appendix A). Liquid chromatography mass spectrometry (LCMS) showed that this fraction contained two primary peaks in both the UV/Vis and base peak chromatograms, exhibiting molecular ions at 305.2023 (**1**) and 344.2129 (**2**) *m*/*z* (Appendix A). Dereplication using common databases failed to yield hits to known compounds; therefore, *C. omnivescoria* EM610 was grown on a large scale (3 × 16 L) in liquid culture. The crude extract was then processed by flash chromatography using C8 resin, yielding five reduced-complexity fractions. LR-LCMS analysis of the five fractions revealed that the non-polar (100% CH_3_CN) fraction contained (**1**) and (**2**) (Appendix A), which were then purified by HPLC (Appendix A).

The molecular formula of **1** was determined to be C_21_H_24_N_2_^+^ (305.2023 *m*/*z*) based on ESI-q-ToF data, indicating that the compound was severely hydrogen-deficient and required 11 degrees of unsaturation (DoU). At high concentration (8 mg/mL), the ^1^H NMR spectrum of **1** exhibited significant line broadening. This was slightly improved by diluting the sample (2 mg/mL) and fuming with trifluoroacetic acid (TFA) vapor before acquiring all 2D NMR data, which significantly reduced the line broadening (Appendix A). Acquisition of a 2D NMR dataset (^1^H,^13^C-HSQC, COSY, ^1^H,^13^C-HMBC, and ^1^H,^15^N-HMBC) confirmed that **1** possesses symmetry, as the HSQC spectra revealed only seven protonated carbons with no detectable exchangeable protons (Appendix A; Table 1).

Detailed analysis of the 2D NMR data, particularly the COSY and HMBC correlations, led to the elucidation of the structure of **1** (Figure 2). A strong COSY correlation was observed between the two methylene groups, H-8/H-8′ (δ_H_ 4.30) and H-7/H-7′ (δ_H_ 2.99), indicating they were adjacent to one another. In addition, H-7/H-7′ had additional strong ^1^H,^13^C-HMBC correlations with the quaternary carbon that was part of the phenyl ring, C-1/C-1′ (δ_C_ 136.9). This established two phenethyl moieties, which tandem MS confirmed via key fragment ions at 105.0664 and 201.1295 *m*/*z* (Figure 2 and Appendix A), thereby accounting for 8 of the 11 DoUs. Analysis of the ^1^H,^13^C-HMBC and ^1^H,^15^N-HMBC correlations for H-8/H-8′ revealed that it exhibited several correlations, including connectivity to the aromatic nitrogen, N-9/N-9′ (δ_N_ 279.9), to the downfield methine (C-11; δ_C_ 134.9), and the last remaining quaternary carbon, C-10/C-10′ (δ_C_ 126.6). Both the ^1^H and ^13^C chemical shifts of C-8/C-8′ support the direct attachment of the methylene to the nitrogen. Several other protons had correlation to N-9, including the singlet vinyl methyl groups (H-12/H-12′; δ_H_ 2.06) and the downfield methine (H-11; δ_H_ 8.90). Considering these correlations and the remaining 3 DoUs, the data supported the presence of an imidazolium heterocycle as shown in Figure 1.

Compound **2** was similarly highly hydrogen-deficient, with a molecular formula of C_23_H_25_N_3_^+^ (344.2129 *m*/*z*), thus requiring 13 DoUs. The ^1^H NMR spectra of **2** showed similar peak broadening, even after treatment with TFA (Appendix A; Table 2). Comparison of the spectra to **1** indicated that **2** lacked symmetry, as evident by the presence of significantly more aromatic signals. Analysis of the tandem MS spectrum revealed two fragment ions of 144.0769 and 201.1295 *m*/*z* (Appendix A), with the latter matching the fragment observed in **1**, which represents the imidazolium and one of the phenethyl groups (Figure 2). A fragment ion of 144.0769 potentially represents a tryptamine moiety, which would account for the increase of 2 DoUs compared to **1**. Analysis of the ^1^H and ^13^C NMR chemical shifts present in **2** but absent in the spectrum of **1**, matched literature precedent for indole. Detailed analysis of the 2D NMR dataset confirmed that one of the phenyl groups in **1** has been replaced with an indole group. We initially characterized these as unknown compounds, but subsequent comparison with published data confirmed their structures as the known bacillimidazoles A and E [26]. Unfortunately, at the time we characterized these compounds, the bacillimidazoles were not cataloged in any of the commonly searched databases (NPAtlas and SciFinder); however, they have since been added to SciFinder.

Imidazolium-containing alkaloids have been found in chemical extracts from various bacterial species, including Flavobacteria (discolins), Bacilli (bacillimidazoles), and Gammaproteobacteria (shewazoles) [26,27,28]. All of these metabolites contain the imidazolium core but differ in their functionalization. Yan and colleagues were the first to propose that this family of metabolites is likely produced via a non-enzymatic pathway [26]. Heavy isotope labeling confirmed that the dimethyl olefin is derived from glucose, probably originating from 2,3-butanedione, a known product of the acetolactate catabolic pathway. Various amines can react spontaneously with 2,3-butanedione to form a diimine, which then reacts with different carboxylic acids and reducing agents to generate the imidazolium core. Bioinformatic analysis of *Bacillus* sp. WMMC1349 did not reveal a plausible biosynthetic gene cluster; however, this strain lacked diacetyl reductase or butanediol dehydrogenases, enzymes that convert 2,3-butanedione into acetoin [26]. The absence of these genes likely leads to an intracellular buildup of 2,3-butanedione. Both the discolin- and bacillimidazole-producing strains (*Tenacibaculum discolor* sv11 and *Bacillus* sp. WMMC1349, respectively) contain an aromatic-L-amino acid decarboxylase, which converts tryptophan and phenylalanine into tryptamine and phenethylamine, respectively [26,27,29]. Genomic analysis of *T. discolor* sv11 identified the *disA* gene as a putative aromatic amino acid decarboxylase [27]. The activity of DisA in *T. discolor* sv11 was confirmed through in vitro experiments. More recently, Ham and colleagues confirmed that much of this process occurs spontaneously [29]. Amine, 2,3-butanedione, and formaldehyde were mixed in water for 24 h, and the targeted products were detected by LCMS.

Mining of the *C. omnivescoria* EM610 genome revealed the presence of an aromatic amino acid decarboxylase, a proposed enzyme that would generate the biogenic amines present in **1** and **2** (Appendix A). Additional mining indicated that *C. omnivescoria* EM610 contains both acetolactate synthase (AS) subunits necessary to synthesize 2-acetolactic acid, but lacks both an acetolactase decarboxylase to convert 2-acetolactic acid to acetoin, and a butanediol dehydrogenase (BDH) to convert 2,3-butanediol to acetoin (Figure 3A). However, formation of 2,3-butanedione from 2-acetolactic acid is also known to occur spontaneously [30]. The AS genes are between another gene cassette that encodes a dihydroxy-acid dehydratase and the ketol-acid reductoisomerase, which together convert 2,3-dihydroxyisovalerate to 2-acetolactate. The *C. omnivescoria* EM610 genome also contains two copies of acetoin (diacetyl) reductase (AR), which converts 2,3-butanedione first to acetoin then to 2,3-butanediol. The significant genomic distance (>1,000,000 bp) between the AR and AS genes suggests that distinct regulators control the expression of the enzymes. Further, both AR enzymes are in proximity (<3500 bp) to the transcriptional regulators GntR and AraC, suggesting that the enzymes are expressed only in the presence of specific metabolites. Together, this indicates that *C. omnivescoria* EM610 likely produces **1** and **2** via the spontaneous conversion of acetolactate to 2,3-butanedione.

To confirm that **1** is produced through non-enzymatic reactions, phenethylamine was incubated with 2,3-butanedione in phosphate-buffered saline (PBS) at room temperature. Immediately upon addition of the 2,3-butanedione to the solution, the mixture turned a dark brown color, and precipitation formed. LCMS analysis of the reaction mixture showed that large amounts of **1** were present even in the absence of formaldehyde (Appendix A). However, it has previously been reported that 2,3-butanedione is relatively unstable and degrades into several compounds, including formaldehyde and methylglyoxal, under normal atmospheric conditions [31]. ^1^H NMR analysis of a fresh bottle of 2,3-butanedione confirms the presence of methylene glycol and formaldehyde in a 40:1:0.03 ratio in CDCl_3_ (Appendix A). A similar study of freshly distilled 2,3-butanedione showed no detectable formaldehyde (Appendix A). In addition, ^1^H NMR monitoring of the reaction every 5 min in D_2_O revealed that the spontaneous reaction is rapid, with significant consumption of 2,3-butanedione within 60 min (Appendix A) and the formation of **1** detected within 5 min upon addition of the amine (Appendix A). The reaction proceeds well when conducted under an inert atmosphere (Appendix A) and in the absence of light (Appendix A). However, the solution must be relatively basic for the reactions to proceed, as no product was detected under acidic conditions (Figure 3B). Finally, we performed reactions with a diverse array of amines, using either 2,3-butanedione alone or with butanoic acid or acetic acid, to generate many of the reported imidazolium-containing alkaloids (Appendix A). The targeted product was detected in 74% of the reactions (Table 3; Appendix A), suggesting that all products are likely produced spontaneously.

Although **1** and **2** are produced spontaneously, we wanted to analyze individual moon snail egg collars for their presence. Individual egg collars were collected from four distinct sample locations across Matlacha Pass Aquatic Preserve in southwest Florida during the winter of 2022. Two of the sites (sites 2 and 3) were in mangrove bays, but on opposing sides of the pass, one site (site 1) was on a sandbar, and the last site (site 4) was a beach off the north tip of Bokeelia (Figure 4A). Portions of individual egg collars were extracted and analyzed by HR-LCMS (Figure 4B). Overall, **1** was detected in 62% of the samples (Appendix A). Peaks were confirmed through retention time matching with an authentic standard and through tandem MS, where diagnostic fragment ions were detected (105.0704 and 201.1392 *m*/*z*). The presence of **1** and **2** in individual egg collar extracts confirms they are being produced in an ecological context. The prevalence of these compounds across the Matlacha Pass suggests they might be involved in the chemical ecology of the microbiomes associated with these egg collars.

The imidazolium-containing alkaloids have been reported to possess a wide range of biological activities, including antimicrobial, antibiofilm, and antiviral [28,32,33]. The bacillimidazoles and discolins have been shown to have antimicrobial, cytotoxic, and anti-inflammatory properties [27,29,34,35]. Several reports show that they exhibit moderate antimicrobial activity, with minimum inhibitory concentrations in the low μg/mL against a range of Gram-positive bacteria, including *Mycobacterium smegmatis*, *Listeria monocytogenes*, *Bacillus subtilis*, and methicillin-resistant *Staphylococcus aureus* (MRSA) [27,29,34,35]. However, this activity appears to be variable, as other laboratories observed no growth inhibition when evaluating the same compounds against closely related pathogens (e.g., *S. aureus*) [26]. We hypothesized that the activity may depend on other physiological properties, such as pH, ion concentrations, and metal levels. Supporting this hypothesis, of the 21 phenethylamine-containing imidazolium alkaloids synthesized by Khatua et al., only those with bromide counterions exhibited antimicrobial activity against *B. subtilis*, whereas those with iodide counterions were inactive [35]. Our results show similar inconsistency in activity. Both **1** and **2** were isolated based on intruder and spot-on-lawn assays on crude mixtures against ecologically relevant pathogens. Compounds **1** and **2** were the most abundant metabolites in the fraction, yet they were relatively inactive in antimicrobial assays against those environmental isolates and other pathogens (Appendix A). Evaluation of **1** and **2** against *C. albicans*, *M. smegmatis*, *L. monocytogenes*, *E. coli*, *B. cereus*, *B. subtilis*, *S. aureus*, and MRSA, under the same conditions as in Wang et al., revealed only ~20% growth inhibition against *B. cereus*, *E. coli*, *L. monocytogenes*, *M. smegmatis*, and *S. aureus*, and ~30% growth inhibition against *B. cereus*, *B. subtilis*, *E. coli*, *L. monocytogenes*, and *S. aureus* for **1** and **2**, respectively, at 64 μg/mL, the highest concentration evaluated (Appendix A). It is possible that the counterion associated with imidazolium alkaloids is essential for activity and that its composition changes during isolation, thereby altering the observed potency. However, under laboratory conditions, **1** is produced at high concentrations (300–350 μg/L), which could allow for high localized concentrations within egg collars. Therefore, the imidazolium-containing alkaloids may serve as antimicrobial agents under appropriate conditions.

## 3. Materials and Methods

*General Experimental Procedures:* All bacterial work was carried out in a SterilGardIII Advance sterile hood. All solvents were of HPLC quality. Crude extracts were semi-purified via flash chromatography on a Biotage Selekt system. LR-LCMS data was obtained using an Agilent 1200 series HPLC system equipped with a photo-diode array detector and a Thermo LTQ mass spectrometer or an Agilent 1100 series HPLC system equipped with a photo-diode array detector and an Agilent 6150 quadrupole mass spectrometer. HPLC purifications were carried out using Agilent 1200 Series or 1260 Infinity II HPLC systems equipped with a photodiode-array detector. HR-ESIMS and HR-MSMS were carried out using an Agilent 6530 q-ToF equipped with a 1290 Infinity II UPLC system or a Thermo Orbitrap Exploris 120 Mass Spectrometer. NMR spectra were recorded in *d_6_*-DMSO, CDCl_3_, or D_2_O with the residual solvent peak as an internal standard (*d_6_*-DMSO: δ_C_ 39.5, δ_H_ 2.50; CDCl_3_: δ_C_ 77.0, δ_H_ 7.26; D_2_O: δ_H_ 4.79) on a Bruker Avance III 600 MHz spectrometer equipped with a triple resonance inverse (CP-TCI) Prodigy N_2_ cooled CryoProbe (^1^H 600 MHz; ^13^C 150 MHz), a Bruker Avance II 500 MHz spectrometer equipped with a CPBBO Prodigy N_2_ cooled CryoProbe (^1^H 500 MHz; ^13^C 125 MHz), or Bruker Neo NMR 500 MHz spectrometer equipped with a H/F C/N TCI 5 mm Prodigy CryoProbe (^1^H 500 MHz; ^13^C 125 MHz). Microplate readings were taken on a BioTek Cytation 3 Cell Imaging Multi-Mode Reader.

*General Bacterial Culturing:* Environmental isolates (moon snail egg collar associated) were grown in Yeast Extract Malt Extract (YEME) media (4 g/L yeast extract, 10 g/L malt extract, 4 g/L dextrose) in artificial seawater (SW). Human pathogens (*Escherichia coli*, *Pseudomonas aeruginosa*, *Staphylococcus aureus*, methicillin-resistant *S. aureus* (MRSA), *Listeria monocytogenes*, and *Bacillus cereus*) were grown in Luria–Bertani (LB) media (10 g/L tryptone, 5 g/L yeast extract, 10 g/L NaCl); *Candida albicans* was grown in Yeast Peptone Dextrose (YPD) media (10 g/L yeast extract, 20 g/L peptone, 20 g/L dextrose). The model organism *M. smegmatis* was grown on Middlebrook 7H9 agar plates (0.5 g/L NH_4_SO_4_, 0.5 g/L L-glutamic acid, 0.1 g/L Na-citrate, 1 mg/L pyridoxine, 0.5 mg/L biotin, 2.5 g/L Na_2_HPO_4_, 1.0 g/L KH_2_PO_4_, 40 mg/L ferric ammonium citrate, 50 mg/L MgSO_4_, 0.5 mg/L CaCl_2_, 1 mg/L ZnSO_4_, 1 mg/L CuSO_4_, 15 g/L agar) supplemented with 0.2% (*v*/*v*) glycerol and 0.05% (*v*/*v*) Tween 80 or in Brain Heart Infusion (BHI) broth (6 g/L brain heart, infusion from (solids), 6 g/L peptic digest of animal tissue, 5 g/L NaCl, 3 g/L dextrose, 14.5 g/L pancreatic digest of gelatin, 2.5 g/L Na_2_HPO_4_). *B. subtilis* was grown in LB. Ca^2+^ and Mg^2+^ were adjusted in Mueller-Hinton broth (17.5 g/L casein acid hydrolysate, 2 g/L beef extract, 1.5 g/L starch) to 21.76 mg/L and 11.12 mg/L, respectively.

*Intruder Competition Assays:* “Resident” bacteria were inoculated in 5 mL of YEME + SW at 30 °C with shaking at 175 rpm for 16 h. An aliquot (5 µL) of each resident was introduced to the middle of each well in a 6-well plate containing 10 mL ISP2 (YEME + 18 g/L agar) + SW media. Plates were incubated at 30 °C for at least 16 h or until the colony was well established in a static incubator. “Intruder” bacteria were started in 5 mL cultures as described above for the resident strains. Sterile cotton swabs were used to transfer “intruder” bacteria from the overnight cultures to each well containing a resident. Plates were incubated at 30 °C in a static incubator until the growth morphology of the “intruder” strain was clearly observed, and for a minimum of 16 h. The growth morphology of all intruder assays was documented via photograph, and the zones of inhibition were measured. All intruder assays were performed in duplicate.

*Large Scale Cultivation and Extraction of C. omnivescoria EM610:* An aliquot (500 μL) of an overnight culture (5 mL) of *C. omnivescoria* EM610 was used to inoculate a 50 mL culture, which was incubated at 30 °C for 16 h in a shaker (150 rpm). Once the culture was dense, 10 mL was used to inoculate 1 L of YEME + SW containing hydrophobic resins [XAD4 (5 g/L), XAD7 (5 g/L), and HP20 (10 g/L)]. Cultures were incubated at 30 °C for 5 d, with shaking at 170 rpm. The resin from a 1 L culture was vacuum filtered through miracloth and extracted with acetone and MeOH for 4 and 16 h, respectively. The organic extract was concentrated to 5 mL, loaded onto a solid phase extraction C18 column and eluted in a stepwise gradient to generate the following fractions: 100% water (fraction A), 50% water/50% MeOH (fraction B), and 100% MeOH (fraction C). These extracts were used in the *Spot-on-Lawn Antimicrobial Assays* methods section. Large scale cultures were generated as described above. The resins from 16 L were vacuum filtered using miracloth, and extracted with acetone and MeOH for 4 and 16 h, respectively, at rt. The organic extracts were combined and dried on 8 g of celite under vacuum, yielding a crude extract. This process (growth through extraction) was repeated three times. The crude extracts were semi-purified via Biotage flash chromatography using 8 g Phenomenex Supra C8 resin (50 µm, 65 Å). Using a step gradient, collecting 60 mL of each solvent: 100% H_2_O (fraction A), 75% H_2_O/25% CH_3_CN (fraction B), 50% H_2_O/50% CH_3_CN (fraction C), 25% H_2_O/75% CH_3_CN (fraction D), and 100% CH_3_CN (fraction E). Individual fractions were dried under vacuum.

*Spot-on-Lawn Antimicrobial Assays:* Assays were performed with environmental isolates *Sufflavibacter* sp. EM538 and *Zunongwangia* sp. EM537. Liquid cultures (5 mL) of both EM537 and EM538 were inoculated from single colonies in YEME + SW and placed in a shaking incubator at 175 rpm at 30 °C for 16 h. An aliquot (40 μL) of the overnight culture was added to 10 mL of liquid top agar (YEME + SW with 10 g/L agar) and poured onto an ISP2 + SW plate. Semi-crude fractions and pure compounds were resuspended in 50% MeOH/50% H_2_O at 50 mg/mL, and 2 μL of the solution was placed on top of the top agar. Spots were allowed to dry completely before plates were transferred to stationary incubators at 30 °C. Plates were monitored until a lawn of bacteria had formed and then analyzed for zone of inhibition (growth inhibition).

*Purification of Bacillimidazol A (***1***) and E (***2***) from Cellulophaga sp. EM610 Large Scale Cultivation:* Fraction C exhibited growth inhibition in the spot-on-lawn assays, so individual components were purified by RP-HPLC. Using an Agilent 1200 series HPLC equipped with a Phenomenex 5 μm Luna Phenyl-Hexyl column (250 × 10 mm, 100 Å) with DAD detector using the following solvent system: holding 35% CH_3_CN + 0.1% formic acid (FA)/65% H_2_O + 0.1% FA for 5 min, then gradient to 38.6% CH_3_CN + 0.1% FA/61.4% H_2_O + 0.1% FA over 13 min, followed by a gradient to 100% CH_3_CN + 0.1% FA over 2 min, finally holding 100% CH_3_CN + 0.1% FA for 5 min with a flow rate 3 mL/min. Processing of 48 L of crude extract led to the isolation of 15 mg of **1** (t_R_ = 12.7 min), and 8 mg of **2** (t_R_ = 13.4 min).

*Bacillimidazole A (***1***):* yellow/brown amorphous solid; ^1^H NMR (*d*_6_-DMSO, 500 MHz): δ 8.86 (s, 1H), 7.31 (t, 4H, J = 7.5), 7.26 (t, 2H, J = 7.5), 7.15 (d, 4H, J = 7.5), 4.30 (t, 4H, J = 7.2), 2.99 (t, 4H, J = 7.2), 2.06 (s, 6H); ^13^C{1H} NMR (*d*_6_-DMSO, 150 MHz) δ 136.9 × 2, 134.9, 129.0 × 4, 128.8 × 4, 127.4 × 2, 126.6 × 2, 47.5 × 2, 35.4 × 2, 7.7 × 2; ^15^N NMR (extrapolated from ^1^H,^15^N-HMBC; *d*_6_-DMSO): δ 279.9 × 2; ESI MS/MS (q-ToF) *m*/*z*: 305.1985 (C_21_H_25_N_2_), 201.1295 (C_13_H_17_N_2_), 105.0664 (C_7_H_7_N); HRMS (ESI) *m*/*z*: [M]^+^ Calcd for C_21_H_25_N_2_^+^ 305.2012; Found 305.2023, Δ 3.6.

*Bacillimidazole E (***2***):* yellow/brown amorphous solid; ^1^H NMR (*d*_6_-DMSO, 500 MHz): δ 10.96 (bs, 1H), 8.79 (s, 1H), 7.43 (d, 1H, J = 7.6), 7.36 (d, 1H, J = 7.8), 7.29 (m, 2H), 7.24 (m, 1H), 7.15 (m, 2H), 7.11 (m, 1H), 7.08 (m, 1H), 6.99 (m, 1H), 4.30 (m, 2H), 4.23 (t, 2H, J = 7.2), 3.11 (t, 2H, J = 6.9), 2.99 (t, 2H, J = 7.2), 2.12 (s, 3H), 2.05 (s, 3H); ^3^C{1H} NMR (*d*_6_-DMSO) δ 137.0, 136.4, 134.2, 128.7 × 2, 128.5 × 2, 127.0, 126.8, 126.4, 125.5, 123.3, 121.1, 118.4, 117.7, 111.5, 109.1, 47.1, 47.05, 35.1, 25.4, 7.3, 7.1; ESI MS/MS (q-ToF) *m*/*z*: 344.2088 (C_23_H_26_N_3_), 201.1295 (C_13_H_17_N_2_), 144.0769 (C_10_H_10_N), 105.0666 (C_7_H_7_N); HRMS (ESI) *m*/*z*: [M]^+^ Calcd for C_23_H_26_N_3_^+^ 344.2121; Found 344.21286, Δ 2.3.

*Genome Sequencing, Assembly, and Genome Mining of Cellulophaga omnivescoria EM610:* A single colony of *C. omnivescoria* EM610 was grown in 5 mL liquid culture to an OD_600_ of 2 (~10^9^ cells). Cells were harvested via centrifugation (4000 rpm for 10 min at 4 °C), the supernatant was decanted, and the cell pellet was frozen at −20 °C. The cell pellet was sent to SeqCenter (Pittsburgh, PA, USA) for DNA extraction, sequencing, and assembly. DNA was sequenced using 600 Mbp Nanopore technology on an Oxford Nanopore sequencer. Residual adapter sequences from Oxford Nanopore Technology (ONT) missed during base calling and demultiplexing were trimmed with porechop. De novo genome assemblies were generated with Flye6 using the nano-hq (ONT high-quality reads) model. Assembled contigs were evaluated for circularization via Circulator and the assembled genome was annotated with Bakta8 (using the Bakta v5 database). Assembly statistics were recorded with QUAST (Appendix A). Species-level identification was accomplished by submission of the assembled genome to the Type (Strain) Genome Server (TYGS) (Appendix A). The assembled genome was also uploaded to RAST (Rapid Annotation using Subsystem Technology), and the resulting annotation was downloaded as an excel file. The annotation file for *C. omnivescoria* EM610 was manually examined for the presence of AlsS subunits (2-acetolactate synthase), AlsD (2-acetolactate decarboxylase), DisA (phenylalanine decarboxylase), and butanediol-dehydrogenases. Homologs of these genes were blasted against the *C. omnivescoria* EM610 genome to ensure nothing had been misannotated.

*Egg Collar Collection and Sample Preparation:* Moon snail egg collars belonging to *Neverita delessertiana* were collected from four sites in the Matlacha Pass Aquatic Preserve in SW Florida in January 2023 (site 1: 82°5′ 17.0″ W 26°42′ 12.462″ N, site 2: 82°6′ 37.54″ W 26°40′ 22.638″ N, site 3: 82°4′ 54.1″ W 26°42′ 1.4″ N, and site 4: 82°8′ 15.51″ W 26°42′ 10.662″ N) with permission from the Florida Wildlife Conservation (permit number: SAL-21-2113-SR). For each site, 10 egg collars were collected, placed in a Falcon tube, and stored at −20 °C. Individual egg collars were lyophilized for 24 h, then homogenized with a mortar and pestle, and extracted with acetone (5 mL) for 16 h at rt while shaking at 150 rpm. The organic layer was decanted and further extracted with MeOH (5 mL) for 4 h at rt. The combined organic layers were filtered through a glass pipette column celite plug and concentrated under vacuum. Dried extracts were stored at −20 °C until purification. A subset (16) of extracts (4 from each site) was semi-purified using a glass pipette column containing ~100 mg of C8 resin (Phenomenex Supra C8; 50 µm, 65 Å). Egg collar extracts were resuspended in ~0.5 mL 100% H_2_O and applied to the C8 plug and eluted with 3 mL of 100% H_2_O (fraction A), 75% H_2_O/25% MeOH (fraction B), and 100% MeOH (fraction C). Fractions were dried under vacuum, and then fraction C was analyzed by HR-LCMS using the following method. Using a Thermo Vanquish UPLC equipped with a Phenomenex 2 μm Kinetex Polar C18 column (100 × 2.1 mm, 100 Å) with Orbitrap Exploris 120 detector using the following solvent system: holding 5% CH_3_CN + 0.1% FA/95% H_2_O + 0.1% FA for 1 min then gradient to 25% CH_3_CN + 0.1% FA/75% H_2_O + 0.1% FA over 0.1 min, followed by a gradient to 100% CH_3_CN + 0.1% FA over 6.9 min holding 100% CH_3_CN + 0.1% FA for 2.5 min with a flow rate 0.5 mL/min. MS data were collected using a Thermo Exploris 120 in positive mode using Thermo Freestyle software. Electrospray ionization parameters were set to 50 L/min for sheath gas, 10 L/min for auxiliary gas, 1 L/min for sweep gas and 350 °C for auxiliary gas temperature. The spray voltage was set to 3500 V, the ion transfer tube temperature to 325 °C. MS1 data were collected from 150 to 2000 *m*/*z* with a resolution of 60,000 at *m*/*z* 200 with 1 micro scan. The maximum ion injection time was set to auto with an automatic gain control (AGC) target of standard. Tandem MS spectra were collected using data-dependent acquisition, targeted ion list for MS1 ion selection for fragmentation. HCD collision energy was set to 45. MS2 data were collected with a resolution of 15,000 at *m*/*z* 200 with 1 micro scan and an AGC set to standard.

*Non-Enzymatic Synthesis of Bacillimidazoles, Discolins, and other Biogenic Amine-Containing Imidazolium Compounds:* The general method for the synthesis of **1**–**9** involved charging a 10-dram vial with 5 mL of phosphate-buffered saline (PBS) and 2,3-butanedione (430 μL, 5 mmol). Then, the amine (12.5 mmol) was added to the vial; when two amines were added, each was added at 6.25 mmol. The reaction mixture was left shaking rt for 16 h. Reaction mixtures were extracted with EtOAc (3 × 5 mL), the organic layer was concentrated, and an aliquot was analyzed by LR-LCMS. Reactions containing either acetic acid (286 μL, 5 mmol), **1a**–**9a** or butyric acid (458 μL, 5 mmol), **1b**–**9b**, were conducted under similar conditions, except the acid was added after the 2,3-butanediol. Reactions **1**–**3** were repeated with freshly distilled 2,3-butanedione as described, except that the reactions were shaken in the dark. Reactions generating **1a**–**1c**, **2a**–**2c**, **3a**, **3b**, **5**, and **5a** were run on an Agilent 1200 series HPLC system equipped with a photo-diode array detector and a Thermo LTQ mass spectrometer using the following method: holding 10% CH_3_CN + 0.1% FA/90% H_2_O + 0.1% FA for 3 min then gradient to 100% CH_3_CN + 0.1% FA over 10 min at a flow rate of 0.3 mL/min. All other reactions were run on an Agilent 1100 series HPLC system equipped with a photo-diode array detector and an Agilent 6150 quadrupole mass spectrometer using the following method: holding 10% CH_3_CN + 0.1% FA/90% H_2_O + 0.1% FA for 1 min then gradient to 100% CH_3_CN + 0.1% FA over 9 min at a flow rate of 0.3 mL/min. Pseudo-MSMS fragmentation was obtained on the Agilent 6150 quadrupole mass spectrometer by increasing the fragmentation energy to 190 V.

*Varying Light, Oxygen, and pH Conditions of the Non-Enzymatic Reactions:* Reactions with phenethylamine were set up and analyzed similarly as described above, but with the following exceptions. Dark reactions: Prior to charging the vial with any reagent, the vial was wrapped in aluminum foil. The reaction mixture was then added as described and left at rt for 16 h. Oxygen-free reactions: Prior to charging a 50-mL flask with any reagent, the flask was dried under heat and purged with N_2_ (*g*) and kept under an inert atmosphere for the duration of the reaction. pH dependency: To evaluate the effect of pH on the reaction outcome, the vial was first charged with PBS and the amine, then the pH was adjusted to a final pH of 5, 6, 7, 8, 9, and 10 with concentrated HCl (*aq*), at which point, 2,3-butanedione was added to the mixture.

*Kinetics Reaction of the Formation of Bacillimidazole A (***1***):* 2,3-Butanedione (43.5 μL, 0.5 mmol) was added to D_2_O and transferred to an NMR tube. An initial proton NMR spectrum was acquired (8 scans). The NMR tube was removed, and the reaction was initiated by the addition of phenethylamine (78.5 μL, 1.25 mmol). The NMR tube was placed back in the instrument. Protons (8 scans) were acquired every 5 min for 3 h with a 5 s relaxation delay between scans.

*Antimicrobial Assays:* Compounds **1** and **2** were screened for antibacterial activity against the human bacterial pathogens *E. coli*, *S. aureus*, MRSA, *P. aeruginosa*, *L. monocytogenes*, and *B. cereus*, against the tuberculosis model organism *M. smegmatis*, against *B. subtilis*, and for antifungal activity against the human fungal pathogen *C. albicans.* Compounds **1** and **2** were also screened for antibacterial activity against other bacteria isolated from the moon snail egg collars (*Sufflavibacter* sp. EM538, *Zunongwangia* sp. EM537, and *Olleya* sp. EM584). Stock solutions of each compound were prepared in DMSO at 6.4 mg/mL. All cultures were grown in the appropriate media (see *General Bacterial Culturing*) and were diluted to an OD_600_ of 0.001 from overnight cultures in either YEME + SW (environmental isolates), cation-adjusted Mueller-Hinton broth (*L. monocytogenes*, *B. subtilis*, *MRSA*, *M. smegmatis*) or LB (remaining human pathogens). Each compound (2 µL) was transferred to individual wells in a 96-well plate, and 198 µL of either diluted bacterial or fungal cultures was added. MRSA, *B. subtilis*, *L. monocytogenes* and *M. smegmatis* were incubated at 37 °C in a shaking incubator. MRSA and *B. subtilis* were incubated for 24 h, *L. monocytogenes* and *M. smegmatis* were incubated for 48 h (plates for *L. monocytogenes* were read at 18 and 48 h). All other human pathogens were incubated at 37 °C in a static incubator for 24 h. Environmental isolates were incubated at 30 °C in a static incubator until negative control cultures reached the stationary phase (>48 h). Cells were resuspended via pipetting, and cell density was measured via absorbance at 595 nm.

## Figures and Tables

**Figure 1 molecules-31-00159-f001:**
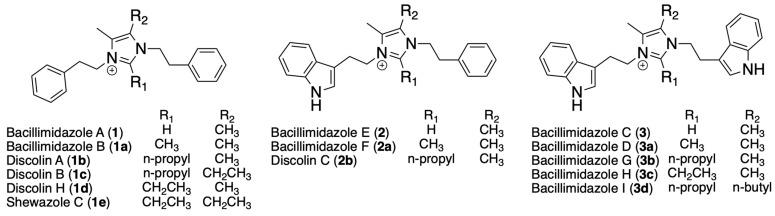
Structures of a subset of imidazolium-containing alkaloids produced by phylogenetically diverse bacteria.

**Figure 2 molecules-31-00159-f002:**
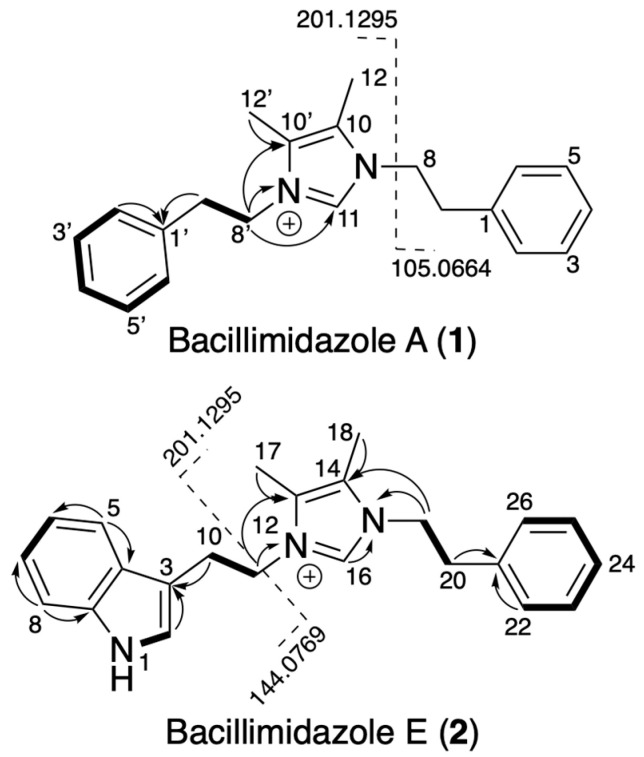
Key 2D NMR and MS/MS fragmentation patterns for bacillimidazoles A (**1**) and E (**2**). Bold lines indicate COSY correlations, and HMBC correlations are indicated by curved arrows.

**Figure 3 molecules-31-00159-f003:**
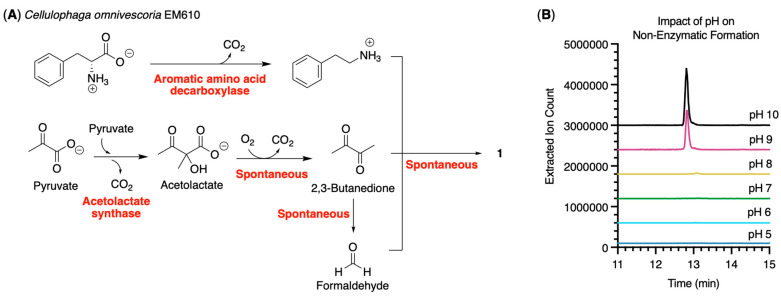
(**A**) Genome mining of *C. omnivescoria* EM610 revealed the presence of an aromatic amino acid decarboxylase (top reaction) and acetolactate synthase (bottom reaction). The absence of acetolactate decarboxylase indicates that the formation of 2,3-butanedione is spontaneous, leading to the spontaneous, non-enzymatic formation of **1** and related compounds (red text denotes enzymatic or spontaneous reactions); and (**B**) Non-enzymatic reactions performed at six different pH values showed that **1** formation occurs only under basic conditions (extracted ion count *m*/*z* 305.2).

**Figure 4 molecules-31-00159-f004:**
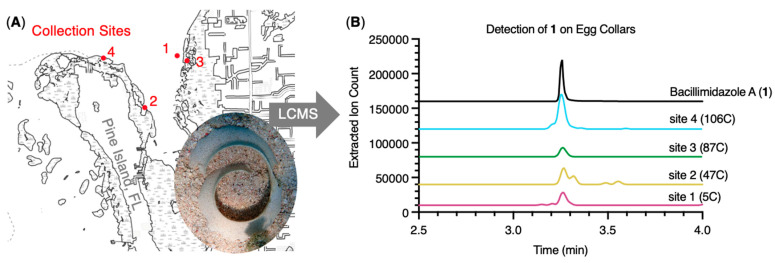
Detection of **1** on the moon snail egg collars. (**A**) Egg collars were collected at four sites across Matlacha Pass Aquatic Preserve in southwest Florida (collection sites indicated in red); and (**B**) Analysis of egg collar extracts by HR-LCMS revealed the presence of **1** in 62% of the egg collars (intensity of 106C was divided by 10 and **1** was divided by 1000) (extracted ion count *m*/*z* 305.2).

**Table 1 molecules-31-00159-t001:** NMR assignments for bacillimidazole A (**1**) in *d*_6_-DMSO.

Position	δ_C_, Type ^a^	δ_H_ (*J* in Hz) ^b^	δ_N_ ^c^	^1^H,^13^C-HMBC ^b^	COSY ^b^	^1^H,^15^N-HMBC ^b^
1,1′	136.9, C					
2,2′	129.0, CH	7.15, d (7.5)		3/3′, 4/4′, 5/5′, 7/7′	3/3′, 4/4′	
3,3′	128.8, CH	7.31, t (7.5)		2/2′, 4/4′, 6/6′	2/2′, 4/4′	
4,4′	127.4, CH	7.26, t (7.5)		2/2′, 3/3′, 5/5′, 6/6′	2/2′, 3/3′, 5/5′, 6/6′	
5,5′	128.8, CH	7.31, t (7.5)		3/3′, 4/4′, 5/5′	4/4′, 6/6′	
6,6′	129.0, CH	7.15, d (7.5)		3/3′, 4/4′, 5/5′, 7/7′	4/4′, 5/5′	
7,7′	35.4, CH_2_	2.99, t (7.2)		1, 6, 8	8/8′	9/9′
8,8′	47.5, CH_2_	4.30, t (7.2)		1, 6, 7, 11	7/7′	9/9′
N9,9′	----		279.9			
10,10′	126.6, C					
11	134.9, CH	8.86, s		8, 10		9/9′
12,12′	7.7, CH_3_	2.06, s		10	12/12′	9/9′

^a^ 125 MHz for ^13^C NMR; ^b^ 600 MHz for ^1^H NMR, ^1^H,^13^C-HMBC, COSY, ^1^H,^15^N-HMBC; ^c^ extrapolated from ^1^H,^15^N-HMBC.

**Table 2 molecules-31-00159-t002:** NMR assignments for bacillimidazole E (**2**) in *d*_6_-DMSO.

Position	δ_C_, type ^a^	δ_H_ (*J* in Hz) ^b^	δ_N_ ^c^	^1^H,^13^C-HMBC ^b^	COSY ^b^	^1^H,^15^N-HMBC ^b^
ΝH1	----	10.96, bs	301.1		2	
2	123.3, CH	7.11, m		3, 9	NH1	N1
3	109.1, C					
4	127.0, C					
5	111.5, CH	7.36, d (7.8)		4, 6	7	
6	118.4, CH	6.99, m		4, 5	7, 8	N1
7	121.1, CH	7.08, m		8, 9	5, 6	
8	117.7, CH	7.43, d (7.6)		7, 9	6	
9	136.4, C					
10	25.4, CH_2_	3.11, t (6.9)		2, 3, 4, 11	11	
11	47.05, CH_2_	4.30, m		3, 10, 13, 16	10	N12
N12	----		332.4			
13	125.5, C					
14	126.4, C					
N15	----		330.1			
16	134.2, CH	8.79, s		11, 13, 14, 19		N12, N15
17	7.1, CH_3_	2.12, s		14		N12
18	7.3, CH_3_	2.05, s		13		N15
19	47.1, CH_2_	4.23, t (7.2)		13, 16, 20, 21	20	N15
20	35.1, CH_2_	2.99, t (7.2)		19, 21, 22	19	
21	137.0, C					
22	128.7, CH	7.15, m		20, 21, 23	23	
23	128.5, CH	7.29, m		22, 24	22	
24	126.8, CH	7.24, m		23, 25		
25	128.5, CH	7.29, m		24, 26	26	
26	128.7, CH	7.15, m		20, 21, 25	25	

^a^ 125 MHz for ^13^C NMR; ^b^ 600 MHz for ^1^H NMR, ^1^H,^13^C-HMBC, COSY, ^1^H,^15^N-HMBC; ^c^ extrapolated from ^1^H,^15^N-HMBC.

**Table 3 molecules-31-00159-t003:** Outcome of non-enzymatic reactions to form diverse imidazolium-containing alkaloids.

Amine	2,3-Butanedione	Acetic Acid	Butyric Acid	Product (*m*/*z*)	Product Detected
Phenethylamine	X			**1** (305)	Yes
	X	X		**1a** (319)	Yes
	X		X	**1b** (347)	ND
Phenethylamine and tryptamine	X			**2** (344)	Yes
	X	X		**2a** (358)	Yes
	X		X	**2b** (386)	Yes
Tryptamine	X			**3** (383)	Yes
	X	X		**3a** (397)	Yes
	X		X	**3b** (425)	ND
Tyramine	X			**4** (337)	Yes
	X	X		**4a** (351)	Yes
	X		X	**4b** (379)	ND
Benzylamine	X			**5** (277)	Yes
	X	X		**5a** (291)	Yes
	X		X	**5b** (319)	ND
Isobutyl amine	X			**6** (209)	Yes
	X	X		**6a** (223)	Yes
	X		X	**6b** (251)	Yes
Phenethylamine and tyramine	X			**7** (321)	Yes
	X	X		**7a** (335)	Yes
	X		X	**7b** (363)	ND
Tryptamine and tyramine	X			**8** (360)	Yes
	X	X		**8a** (374)	Yes
	X		X	**8b** (402)	Yes
Isobutyl amine and tryptamine	X			**9** (296)	ND
	X	X		**9a** (310)	Yes
	X		X	**9b** (338)	ND

ND = not detected.

## Data Availability

The data resulting from this study are available in the published article and its Appendix A. NMR data is available at NP-MRD under accession numbers NP0351890 and NP0351891 at (https://np-mrd.org/, accessed on 15 November 2025). *Cellulophaga omnivescoria* EM610 genome is available at NCBI under BioProject ID PRJNA1355476.

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
