# Peer review of "Ecological Prevalence and Non-Enzymatic Formation of Imidazolium Alkaloids on Moon Snail Egg Collars"

_molecules, 2026, doi:10.3390/molecules31010159_

Round 1
Reviewer 1 Report
Comments and Suggestions for Authors
This manuscript by Mevers and coworkers describes the isolation and structure characterization of two known imidazolium metabolites, bacillimidazole A and E, from the moon snail Cellulophaga omnivescoria EM610. The authors further support the nonenzymatic formation of these imidazolium analogs through biomimetic synthesis. This manuscript is well-written and scientifically sound. Therefore, I believe the manuscript could be suitable for publication in Molecules after minor revision as below.
Specific comments:
- P. 3, line 85-99. Bacillimidazole A and E were isolated by activity-guided isolation, yet both were inactive. Based on the LCMS chromatogram (Figure S3), it appears that other compounds present in the fraction may exhibit antibiotic activity. Notaboly, there is another major peak at ~18 min in the crude extract. Please provide an explanation regarding this observation and whether additional compounds might contribute to the observed activity.
- P. 4, line 141-143. The authors state that bacillimidazole A and E are not cataloged in SciFinder; however, both compounds can be found when searched using either chemical structure or SMILES representations. It is likely that the SciFinder database has been updated. Please revise this statement accordingly.
SMILES for Bacillimidazole A: CC1=C(C)[N+](CCC2=CC=CC=C2)=CN1CCC3=CC=CC=C3
Author Response
Comments and Suggestions for Authors
This manuscript by Mevers and coworkers describes the isolation and structure characterization of two known imidazolium metabolites, bacillimidazole A and E, from the moon snail Cellulophaga omnivescoria EM610. The authors further support the nonenzymatic formation of these imidazolium analogs through biomimetic synthesis. This manuscript is well-written and scientifically sound. Therefore, I believe the manuscript could be suitable for publication in Molecules after minor revision as below.
Specific comments:
- P. 3, line 85-99. Bacillimidazole A and E were isolated by activity-guided isolation, yet both were inactive. Based on the LCMS chromatogram (Figure S3), it appears that other compounds present in the fraction may exhibit antibiotic activity. Notaboly, there is another major peak at ~18 min in the crude extract. Please provide an explanation regarding this observation and whether additional compounds might contribute to the observed activity.
Response: The peak at 18 min in the chromatogram represents column bleed. It is a synthetic compound that is part of another project within the lab. It was our mistake not to indicate this in the original submission. We have added a chromatogram that identifies this other peak. We did test all other HPLC fractions, but we lost activity in the spot-on-lawn assay at this stage of the project. Reevaluation of 1 and 2 at higher concentrations (64 μg/mL) against the same pathogens used in the previous papers indicates that both compounds can inhibit about 40% of the growth of Gram-positive bacteria at this concentration.
- P. 4, line 141-143. The authors state that bacillimidazole A and E are not cataloged in SciFinder; however, both compounds can be found when searched using either chemical structure or SMILES representations. It is likely that the SciFinder database has been updated. Please revise this statement accordingly.
SMILES for Bacillimidazole A: CC1=C(C)[N+](CCC2=CC=CC=C2)=CN1CCC3=CC=CC=C3
Response: We thank the reviewer for pointing this out. We have updated the manuscript accordingly.
Reviewer 2 Report
Comments and Suggestions for Authors
This manuscript focuses on the chemical interactions among marine microorganisms in Neverita delessertiana egg collars. Using an ecology-driven screening strategy, it identifies two natural products with potential ecological functions and reveals their likely non-enzymatic synthesis mechanism. However, the innovation of the manuscript is relatively weak, and some conclusions warrant further discussion. The specific revision suggestions are as follows:
- Based on the NMR spectra provided in the supplementary materials, the purity of compounds 1 and 2 appears poor, which affects the accuracy of both the structural determination and the subsequent bioactivity assay results. Please provide the purity analysis of compounds.
- Both isolated compounds are previously reported compounds, whose chemical structures have already been thoroughly and rigorously elucidated. The chemical novelty of manuscript is weak.
- Compounds 1 and 2 were isolated based on inhibitory activity and were the most abundant metabolites in the active fraction. And they are widely present in moon snail egg collars. However, they showed no activity in subsequent standardized antimicrobial assays. How can the role of these compounds in microbial competition be explained?
Author Response
Comments and Suggestions for Authors
This manuscript focuses on the chemical interactions among marine microorganisms in Neverita delessertiana egg collars. Using an ecology-driven screening strategy, it identifies two natural products with potential ecological functions and reveals their likely non-enzymatic synthesis mechanism. However, the innovation of the manuscript is relatively weak, and some conclusions warrant further discussion. The specific revision suggestions are as follows:
- Based on the NMR spectra provided in the supplementary materials, the purity of compounds 1 and 2 appears poor, which affects the accuracy of both the structural determination and the subsequent bioactivity assay results. Please provide the purity analysis of compounds.
Response: The 1H NMR of these compounds suffers from significant line broadening, and this contributes to the quality of the spectra. We believe this is due to pi-stacking between the imidazolium ring and the side chains, which reduces free rotation and causes some signal doubling. In particular, in 2, the phenyl group has 5 non-equivalent protons, arising from intramolecular interactions that prevent free rotation. We have included UV chromatograms of the purified compounds in the supplemental information to demonstrate the samples' purity.
- Both isolated compounds are previously reported compounds, whose chemical structures have already been thoroughly and rigorously elucidated. The chemical novelty of manuscript is weak.
Response: We acknowledge that the chemical novelty of the manuscript is weaker than we had hoped. Unfortunately, we were not aware that the compounds were previously described metabolites until we solved the structures, as they were omitted from all commonly used libraries. However, we do feel that our non-enzymatic experiments add valuable contributions to the current understanding of these systems.
- Compounds 1 and 2 were isolated based on inhibitory activity and were the most abundant metabolites in the active fraction. And they are widely present in moon snail egg collars. However, they showed no activity in subsequent standardized antimicrobial assays. How can the role of these compounds in microbial competition be explained?
Response: We have retested 1 and 2 at higher concentrations (64 μg/mL), and following the methods of the previous papers, including pathogen choice. We observed moderate growth inhibition, ~40% inhibition at 64 μg/mL. Under laboratory growth conditions, 1 and 2 are produced at high concentrations (~300 μg/L ), suggesting they may be present at high localized concentrations under ecological conditions on the egg collars. Additionally, reevaluation of the literature supports our claim about the importance of ions within the media as the synthesis of imidazolium alkaloids by Khatua et al. showed only the imidazolium alkaloids with bromine counterions exhibited antibacterial activity.
Reviewer 3 Report
Comments and Suggestions for Authors
The manuscript investigates the chemical interactions among bacteria within moon snail (Neverita delessertiana) egg collars and the resulting metabolites. It describes the isolation and identification of two known imidazolium alkaloids—bacillimidazole A (1) and E (2)—elucidates their biosynthetic pathway through genomic analysis and in vitro non-enzymatic synthesis, and demonstrates their widespread presence in field-collected egg collars. Regrettably, no corresponding antimicrobial activity was observed for the isolated compounds. The study integrates multidisciplinary approaches from microbiology, chemistry, genomics, and synthetic chemistry, which is commendable. However, before acceptance by Molecules, the following issues warrant attention:
1. Clarity on Compound Novelty and Literature Citation: The manuscript initially gives the impression that compounds 1 and 2 are newly discovered, particularly due to the statement: "bacillimidazoles were not cataloged in any of the commonly-searched databases (NPAtlas and SciFinder)." This sentence is highly misleading and should be removed. The authors should explicitly state in the Results and Discussion section: "We initially characterized these as unknown compounds, but subsequent comparison with published data (Yan et al., 2022) confirmed their structures as the known bacillimidazoles A and E." This is crucial for maintaining academic rigor.
2. In-depth Analysis and Discussion of the Bioactivity Paradox: The study's premise, based on ecology-driven bioassay-guided fractionation, was to identify antimicrobial (chemically defensive) agents. However, the major isolated compounds were inactive. The authors only briefly mention in the discussion that "The activity may depend on other physiological properties," which is insufficient. It is recommended to deepen the discussion by:
o Expanding the Bioassay Models: Consider whether the "false-negative" result is due to limited testing models. It is suggested to supplement tests against other potentially relevant targets, such as activity against Gram-positive bacteria (e.g., Staphylococcus aureus) or ecotoxicity assays against algae or plankton.
o Systematically Exploring Reasons for Activity Loss: A more detailed discussion, supported by literature, should be provided on several possibilities: a) Performing a more refined LC-MS/MS analysis of the crude extract to probe for trace, highly active unknown components; b) Potential synergistic effects between compounds 1 and 2 or with other components; c) The possibility that they are prodrugs activated in vivo upon metabolism; d) Whether their activity is highly dependent on specific conditions of the egg collar microenvironment (e.g., pH, metal ion concentrations) not replicated in standard in vitro assays.
3. Prudent Wording in the Abstract, Conclusion, and Title: The wording in the abstract and conclusion should be more precise to avoid misleading readers. For instance, the phrase in the abstract "led to the identification of bacillimidazole A (1) and E (2)" could imply they are the active constituents. It is recommended to revise the abstract along the following lines to better frame the study's logic and core findings (the non-bioactive compounds, their formation mechanism, and ecological distribution):
"Animal-associated microbiomes are arenas of intense chemical competition. To investigate such interactions within the Flavobacteriaceae-dominated microbiota of Neverita delessertiana (moon snail) egg collars, we conducted systematic binary competition assays. This revealed that Cellulophaga omnivescoria EM610 inhibited the growth of several competitors. Chemical investigation of this strain led to the isolation of bacillimidazoles A (1) and E (2), imidazolium alkaloids previously reported from a marine Bacillus. Notably, we demonstrated that these compounds form spontaneously via condensation of 2,3-butanedione with phenethylamine and/or tryptamine under mild, basic conditions. Genomic analysis of EM610 supported this non-enzymatic route, revealing a metabolic predisposition for accumulating the requisite diketone precursor. Crucially, tandem mass spectrometry analysis of individual egg collar extracts detected bacillimidazole A in 62% of field-collected samples, confirming its production in the native ecological context. Although the pure compounds showed no activity in standard antimicrobial assays, their widespread presence suggests these spontaneously formed metabolites may play a nuanced role in the chemical ecology of the moon snail egg collar microbiome. This study highlights non-enzymatic chemistry as a potential source of bioactive small molecules in host-associated microbial communities."
Furthermore, consider revising the title to more accurately reflect the study's focus, for example: "Spontaneous Formation and Ecological Prevalence of Imidazolium Alkaloids on Moon Snail Egg Collars".
4. Other Technical Issues:
o The resolution of Figures 1-4 appears low.
o In Figure 2, the caption should clearly label the 2D-NMR correlations (e.g., using bold lines for COSY and curved arrows for HMBC).
o The HSQC spectrum for compound 1 is missing from the Supporting Information (SI).
o Consider whether the 13C and 1H NMR data for compounds 1 and 2 (currently in SI Tables S2 and S3) should be moved to the main text, or at least have these tables placed earlier within the SI document for easier reference.
Author Response
Comments and Suggestions for Authors
The manuscript investigates the chemical interactions among bacteria within moon snail (Neverita delessertiana) egg collars and the resulting metabolites. It describes the isolation and identification of two known imidazolium alkaloids—bacillimidazole A (1) and E (2)—elucidates their biosynthetic pathway through genomic analysis and in vitro non-enzymatic synthesis, and demonstrates their widespread presence in field-collected egg collars. Regrettably, no corresponding antimicrobial activity was observed for the isolated compounds. The study integrates multidisciplinary approaches from microbiology, chemistry, genomics, and synthetic chemistry, which is commendable. However, before acceptance by Molecules, the following issues warrant attention:
- Clarity on Compound Novelty and Literature Citation: The manuscript initially gives the impression that compounds 1 and 2 are newly discovered, particularly due to the statement: "bacillimidazoles were not cataloged in any of the commonly-searched databases (NPAtlas and SciFinder)." This sentence is highly misleading and should be removed. The authors should explicitly state in the Results and Discussion section: "We initially characterized these as unknown compounds, but subsequent comparison with published data (Yan et al., 2022) confirmed their structures as the known bacillimidazoles A and E." This is crucial for maintaining academic rigor.
Response: We apologize for the misleading statement. We have added the suggestion, but it represents a rephrasing of our original statement, which referenced Yan et al., 2022. Our purpose in including the statement about cataloging in SciFinder was to provide context for how the dereplication failed and to caution the reader against trusting SciFinder alone for novelty.
- In-depth Analysis and Discussion of the Bioactivity Paradox: The study's premise, based on ecology-driven bioassay-guided fractionation, was to identify antimicrobial (chemically defensive) agents. However, the major isolated compounds were inactive. The authors only briefly mention in the discussion that "The activity may depend on other physiological properties," which is insufficient. It is recommended to deepen the discussion by:
o Expanding the Bioassay Models: Consider whether the "false-negative" result is due to limited testing models. It is suggested to supplement tests against other potentially relevant targets, such as activity against Gram-positive bacteria (e.g., Staphylococcus aureus) or ecotoxicity assays against algae or plankton.
Response: We thank the reviewer for this suggestion. Compounds 1 and 2 were tested against both Gram-positive and Gram-negative bacteria, as well as environmental isolates as indicated in Supplemental Table S6. As part of the revision we screened 1 and 2 against Mycobacterium smegmatis, Listeria monocytogenes, S. aureus (MRSA), and B. subtilis at 64 μg/mL. These other strains were selected because they were used in the reported literature. While only observed minor growth inhibition at the highest concentration, ~40% at 64 μg/mL. We have included this extra data and context to the manuscript.
o Systematically Exploring Reasons for Activity Loss: A more detailed discussion, supported by literature, should be provided on several possibilities: a) Performing a more refined LC-MS/MS analysis of the crude extract to probe for trace, highly active unknown components; b) Potential synergistic effects between compounds 1 and 2 or with other components; c) The possibility that they are prodrugs activated in vivo upon metabolism; d) Whether their activity is highly dependent on specific conditions of the egg collar microenvironment (e.g., pH, metal ion concentrations) not replicated in standard in vitro assays.
Response: We did evaluate all fractions generated during our bioassay-guided isolation and it failed to reveal any other active fraction. We do not feel like we are missing any highly active components. Our spot-on-lawn assays were conducted at 50 mg/mL and the zones were relatively small. We have gone back and re-evaluated 1 and 2 against a larger panel of pathogens, including those originally evaluated in the previous reports. We observed minor activity at the highest concentration with ~40% growth inhibition at 64 μg/mL. However, reevaluation of the literature supports our claim about the importance of physiological ions within the media as the synthesis of imidazolium alkaloids by Khatua et al. showed only the imidazolium alkaloids with bromine counterions exhibited antibacterial activity. We have added this extra context to the manuscript.
- Prudent Wording in the Abstract, Conclusion, and Title: The wording in the abstract and conclusion should be more precise to avoid misleading readers. For instance, the phrase in the abstract "led to the identification of bacillimidazole A (1) and E (2)" could imply they are the active constituents. It is recommended to revise the abstract along the following lines to better frame the study's logic and core findings (the non-bioactive compounds, their formation mechanism, and ecological distribution):
"Animal-associated microbiomes are arenas of intense chemical competition. To investigate such interactions within the Flavobacteriaceae-dominated microbiota of Neverita delessertiana (moon snail) egg collars, we conducted systematic binary competition assays. This revealed that Cellulophaga omnivescoria EM610 inhibited the growth of several competitors. Chemical investigation of this strain led to the isolation of bacillimidazoles A (1) and E (2), imidazolium alkaloids previously reported from a marine Bacillus. Notably, we demonstrated that these compounds form spontaneously via condensation of 2,3-butanedione with phenethylamine and/or tryptamine under mild, basic conditions. Genomic analysis of EM610 supported this non-enzymatic route, revealing a metabolic predisposition for accumulating the requisite diketone precursor. Crucially, tandem mass spectrometry analysis of individual egg collar extracts detected bacillimidazole A in 62% of field-collected samples, confirming its production in the native ecological context. Although the pure compounds showed no activity in standard antimicrobial assays, their widespread presence suggests these spontaneously formed metabolites may play a nuanced role in the chemical ecology of the moon snail egg collar microbiome. This study highlights non-enzymatic chemistry as a potential source of bioactive small molecules in host-associated microbial communities."
Response: We attempted to restructure the abstract and conclusions to reflect the paper's core findings better.
Furthermore, consider revising the title to more accurately reflect the study's focus, for example: "Spontaneous Formation and Ecological Prevalence of Imidazolium Alkaloids on Moon Snail Egg Collars".
Response: We have modified the title to better reflect the study.
New Title: “Ecological Prevalence and Non-Enzymatic Formation of Imidazolium Alkaloids on Moon Snail Egg Collars”
- Other Technical Issues:
o The resolution of Figures 1-4 appears low.
Response: The figures appear to be high resolution on our end, but we have reimported them to ensure high quality. This may be an issue when the file gets converted to pdf.
o In Figure 2, the caption should clearly label the 2D-NMR correlations (e.g., using bold lines for COSY and curved arrows for HMBC).
Response: We have updated the figure caption to denote the 2D NMR correlations
o The HSQC spectrum for compound 1 is missing from the Supporting Information (SI).
Response: We thank the reviewer for pointing out this oversight. We have updated the supporting information to include the HSQC spectrum for 1.
o Consider whether the 13C and 1H NMR data for compounds 1 and 2 (currently in SI Tables S2 and S3) should be moved to the main text, or at least have these tables placed earlier within the SI document for easier reference.
Response: We have moved both NMR data tables for compounds 1 and 2 to the main text.
Round 2
Reviewer 2 Report
Comments and Suggestions for Authors
The authors have addressed the reviewers' comments in the submitted manuscript.